# Auditing Black-Box Prediction Models for Data Minimization Compliance

**Bashir Rastegarpanah**
Boston University
bashir@bu.edu

**Krishna P. Gummadi**
MPI-SWS
gummadi@mpi-sws.org

**Mark Crovella**
Boston University
crovella@bu.edu

## Abstract

In this paper, we focus on auditing *black-box* prediction models for compliance with the GDPR's *data minimization principle*. This principle restricts prediction models to use the minimal information that is necessary for performing the task at hand. Given the challenge of the black-box setting, our key idea is to check if each of the prediction model's input features is individually necessary by assigning it some constant value (i.e., applying a simple imputation) across all prediction instances, and measuring the extent to which the model outcomes would change. We introduce a metric for data minimization that is based on model instability under simple imputations. We extend the applicability of this metric from a finite sample model to a distributional setting by introducing a probabilistic data minimization guarantee, which we derive using a Bayesian approach. Furthermore, we address the auditing problem under a constraint on the number of queries to the prediction system. We formulate the problem of allocating a budget of system queries to feasible simple imputations (for investigating model instability) as a multi-armed bandit framework with probabilistic success metrics. We define two bandit problems for providing a probabilistic data minimization guarantee at a given confidence level: a decision problem given a data minimization level, and a measurement problem given a fixed query budget. We design efficient algorithms for these auditing problems using novel exploration strategies that expand classical bandit strategies. Our experiments with real-world prediction systems show that our auditing algorithms significantly outperform simpler benchmarks in both measurement and decision problems.

## 1 Introduction

Concerns about the widespread use of data-driven prediction models and their growing reliance on personal data of individuals, have led to a number of data protection laws and regulations in recent years [17, 19, 24]. Prominent amongst such regulations is the GDPR (article 5.1.c) [19] that proposes the *data minimization* principle to control the extent to which personal data can be acquired and used by prediction models. Specifically, the data minimization principle states that: *"Personal data shall be adequate, relevant and limited to what is necessary for the purposes for which they are processed."*

Despite considerable public debate about the GDPR and the data minimization principle, to date, only a few works [2, 18] have attempted to *operationalize* (i.e., formally interpret) the legal principle for prediction models. These works have focused on finding operationalizations of data minimization that tie the purpose of data processing to some performance metric such as prediction accuracy (e.g., in recommender systems [2], or in classification [18]). Specifically, they study whether a prediction model can be redesigned to achieve similar prediction performance, while using fewer input features. In the process, these works assume a transparent setting, where the full knowledge of the prediction algorithm, training procedure, and desired performance metrics is available to the party investigating the feasibility of reducing data inputs.

35th Conference on Neural Information Processing Systems (NeurIPS 2021).

When putting data minimization into practice, an important case to consider is an auditing scenario where an outsider (auditor) wants to test a given prediction model for compliance with the data minimization principle at deployment time. As prediction algorithms are often important business assets, a more realistic auditing scenario is one in a black-box setting in which the auditor interacts with the model without access to the model internals.

In this paper, we propose an operational definition of the data minimization principle that allows auditing black-box prediction models for compliance at deployment time. In particular, we assume an auditor who does not have access to how the prediction model works, or to the training data, or to information about the purpose of the prediction model. All the auditor can do is using data points to query a given black-box prediction model with a fixed set of input features, and observe the outcomes. We believe our black-box model setting covers many real-world scenarios, as it only requires the designers of prediction models to allow auditors to query their models with prediction instances.

Our key insight is that the auditor can test whether an input feature is needed by the prediction model, by imputing (i.e., guessing) its value and checking the extent to which the outcomes change (i.e., are unstable) for different prediction instances. Intuitively, if the actual value of an input feature is not needed (i.e., can be replaced with a constant) to arrive at similar (stable) outcomes for most prediction instances, then the use of the feature violates the data minimization principle. Note that our instability-based operatlonalization does not require knowledge of the purpose of the prediction model and is independent of performance metrics. Such an operational definition is particularly important given that it is common for companies who provide personalized services to justify data collection simply as "for improving service" [9].

We show how simple imputations, where the actual values of individual features are replaced with a constant value, can be leveraged as a strategy for limiting data inputs to a prediction system at deployment time. We define a data minimization guarantee that is based on a metric of model instability under different feasible simple imputations. While this guarantee induces a procedure for auditing data minimization assuming a finite sample model, we extend the applicability of our auditing framework in two ways. First, we propose a probabilistic audit that allows the auditor to provide a data minimization guarantee at some confidence level with respect to an underlying data distribution. We achieve this goal by introducing the notion of a probabilistic data minimization guarantee and adopting a Bayesian approach. Second, we address the auditing problem under a constraint on the number of queries to the prediction system and we design auditing algorithms that use a budget of system queries strategically in order to provide a data minimization guarantee.

We cast the problem of allocating a query budget to feasible simple imputations into a multi-armed bandit framework, and we formulate two bandit problems that correspond to different auditing tasks given a fixed confidence level: a decision problem given a data minimization level, and a measurement problem given a fixed query budget. Furthermore, we design efficient algorithms for the above problems using exploration strategies for selecting arms. We propose four heuristic exploration strategies: two strategies inspired by Thompson sampling, and two that are custom for our setting.

Finally, we study the effectiveness of our auditing algorithms using different real-world prediction systems. We build prediction models by applying standard model evaluation and feature selection methods, and use the resulting models to perform black-box audits. Our experiments show that algorithms that exploit the proposed bandit framework significantly outperform simpler benchmarks.

## 2   Related Work

**Data Minimization.** Only a few recent papers have addressed the problem of operationalizing data minimization in prediction systems. Biega et al. [2] propose definitions that are based on recommender systems' performance, and conduct an empirical study to check the original recommender performance can be preserved, while limiting the number of known user ratings. The authors in [18] suggest a formulation based on classification accuracy and study the trade-off between data minimization and satisfying other fairness properties. Clavell et al. [10] interpret data minimization as limiting the use of sensitive personal data and do an experimental analysis that suggests data minimization should not be applied without consideration of other social concerns such as fairness.

**Model Instability in Machine Learning.** The instability of prediction models is studied in both adversarial and non-adversarial settings. The traditional non-adversarial setting is concerned with the

instability of a model under different training data [15, 8]. The adversarial setting studies the effects of data perturbations on the model predictions. Perturbations in *training data* is often called data poisoning and has been studied for various prediction models [23]. The more relevant problem to our setting is however studying the effect of *test data* perturbations, which is known as evasion attacks at test time [3, 16].

A key difference between our instability metric and model instability under evasion attacks is in the type of perturbation that we define. While evasion attacks consider perturbations defined as modifying a data instance inside a fixed-radius ball under a particular norm, we consider perturbations defined as projections induced by simple imputations. Moreover, the evasion attack problem concerns the instability of a model prediction locally at each data point whereas the audit problem is concerned with a global notion of instability (see Section 3).

## 3 Model Instability-based Data Minimization Guarantees

In this section, we first formalize our setting and define a metric for model instability under simple imputations. Then we show how this metric can be leveraged to define a data minimization guarantee.

**Setting.** We consider a prediction model being audited at deployment time for adherence to data minimization principle. The model has a fixed set of inputs variables (features) and an output variable from a discrete domain. We assume a black-box setting, i.e., for any prediction instance, an auditor can query the model by specifying the values of all input features and observe the produced output, while the procedure used for generating the output from the inputs is unknown to the auditor. The auditor's goal then is to measure the level of data minimization satisfied by the prediction model using a limited number of queries. A schematic illustration of this setting is provided in Appendix A.

**Notation.** Formally, let $\hat{Y}_F$ denote a prediction model over a set of input features is $F = \{f_1, \ldots, f_d\}$, where each feature $f_i$ takes values from domain $\mathcal{X}_i$. $\mathcal{X} = \prod_{i=1}^{d} \mathcal{X}_i$ denotes the input space of $\hat{Y}_F$. For any prediction instance $\mathbf{x} \in \mathcal{X}$, $\hat{Y}_F(\mathbf{x}) \in \mathcal{Y}$ denotes the prediction made by $\hat{Y}_F$ for $\mathbf{x}$, and $\mathcal{Y}$ is the discrete set of targets in the output space. We also consider a distribution $\mathcal{P}_\mathcal{X}$ over the input space from which samples (data points) are drawn in deployment. The auditor's challenge is to test whether the the model $\hat{Y}_F$ satisfies data minimization principle over $\mathcal{P}_\mathcal{X}$.

**Assessing the need for individual features with simple imputations.** To assess the need for individual features, we propose using simple imputations. Simple imputation is a procedure that is commonly used for handling missing data, and it works by assigning a constant value to a feature $f_i$ in any prediction instance in which $f_i$ is missing. If an auditor finds that imputing a constant value to an input feature $f_i$ has *no or small* effect on outcomes across different prediction instances, then the auditor can conclude that information about the actual values of $f_i$ are *not needed* by $\hat{Y}_F$ to arrive at prediction outcomes. Our idea is to quantify the instability or changes in prediction outcomes to determine the level or extent to which the data minimization principle has been satisfied or violated.

**A metric for model instability under simple imputations.** We now define a metric for instability of a prediction model, $\hat{Y}_F$, under simple feature imputations. Formally, we define an imputation function $\tau_{f_i,b}$ such that for any prediction instance $\mathbf{x}$, $\tau_{f_j,b}(\mathbf{x})$ returns a vector in which the value of feature $f_i$ is replaced with $b \in \mathcal{X}_i$. Let $I_{\hat{Y}_F}(\mathbf{x}, f_j, b)$ be a binary indicator variable that is 1 if the prediction made by $\hat{Y}_F$ for $\mathbf{x}$ changes after imputing $f_j$ with $b$, or 0 otherwise. Formally,

$$I_{\hat{Y}_F}(\mathbf{x}, f_j, b) = \begin{cases} 1 & \text{if } \hat{Y}_F(\mathbf{x}) \neq \hat{Y}_F(\tau_{f_j,b}(\mathbf{x})) \\ 0 & \text{otherwise} \end{cases} \tag{1}$$

Let $X$ be a random variable that takes on values $\mathbf{x} \in \mathcal{X}$ according to an underlying data distribution $\mathcal{P}_\mathcal{X}$. We define the instability of $\hat{Y}_F$ over $\mathcal{P}_\mathcal{X}$ with respect to feature $f_j$ and imputation value $b$ as:

$$\beta_j^b = \mathbb{E}_{X \sim \mathcal{P}_\mathcal{X}} \left[ I_{\hat{Y}_F}(X, f_j, b) \right] \tag{2}$$

In other words, $\beta_j^b$ denotes the probability that the prediction for a data point drawn randomly from $\mathcal{P}_\mathcal{X}$ changes after imputing $f_j$ by $b$.

**Instability-based data minimization guarantee.** We now define a data minimization guarantee an auditor can offer using our model instability metric $\beta_j^b$. Intuitively, the imputation $b$ that induces

iii

the minimum prediction instability for a feature $f_j$ determines how necessary the feature $f_j$ is for generating the predicted outcomes. So a natural data minimization guarantee can be arrived at by finding the greatest lower bound on the need for each individual feature $f_j$ (which in turn requires finding a feasible imputation for each $f_j$ that induces minimum instability).[1] Formally we define:

**Definition 1** *A prediction model $\hat{Y}_F$ satisfies data minimization at level $\beta$ if there does not exist any feature $f_j \in F$ and any imputation value $b \in \mathcal{X}_j$ such that $\beta_j^b < \beta$. The highest level $\beta$ at which data minimization is satisfied constitutes the best data minimization guarantee an auditor can offer for $\hat{Y}_F$.*

Intuitively, a data minimization guarantee at level $\beta$ ensures that every input feature used by a prediction model is indeed necessary to reach the predictions made for at least a certain fraction, $\beta$, of prediction instances. Note that if a prediction model satisfies data minimization at level $\beta$, it also satisfies data minimization at any level $\beta' < \beta$. In practice, the auditor would be interested in finding the best guarantee, i.e., the largest value of $\beta$ at which a prediction system satisfies data minimization.

## 4    Audit Mechanisms for Data Minimization Guarantees

We now consider the challenge of designing efficient black-box audit mechanisms for producing the data minimization guarantee discussed in the previous section. Notice that we defined model instabilities as the expected value of an indicator random variable over an underlying data distribution $\mathcal{P}_\mathcal{X}$. In practice, however, the auditor can only query the black-box model with a finite number of samples drawn from the distribution to estimate the model instability. We call this set of available query samples (prediction instances) the audit dataset $\mathcal{D}^{Audit}$.

**Population audit.** Given an audit dataset and assuming a *finite sample model*, the model instability with respect to each simple imputation, i.e., each (feature, imputation value) pair, can be computed using the the population mean over all prediction instances in the audit dataset. In particular,

$$\hat{\beta}_j^b = \frac{1}{|\mathcal{D}^{Audit}|} \sum_{\mathbf{x} \in \mathcal{D}^{Audit}} \left[ I_{\hat{Y}_F}(\mathbf{x}, f_j, b) \right] \tag{3}$$

The auditor can then measure the data minimization level by exhaustively searching over all features and imputation values and finding the minimum instability based on the the population mean over all prediction instances in the audit dataset. We call this procedure a *"population audit."*

However, in many practical auditing scenarios, a population audit may not be feasible because the number of queries an auditor can issue to the prediction system is limited, and an exhaustive search would exceed the query budget many times over. Furthermore, auditors are often interested in a guarantee that is valid over yet unseen samples drawn from the underlying data distribution.

In the following, we address the above shortcomings of a population audit by introducing a probabilistic audit mechanism for data minimization. In a probabilistic audit, the auditor investigates the prediction model's instability by observing its outputs for a limited number of query data points, and offers a probabilistic guarantee about the data minimization level satisfied by the model.

**Probabilistic data minimization guarantee.** In order to design a probabilistic audit mechanism, we first define the probabilistic data minimization guarantee. This guarantee allows an auditor to extend the applicability of the instability-based data minimization metric from a finite sample model to a distributional setting. Similarly to the data minimization guarantee in Definition 1, we define:

**Definition 2** *A prediction model $\hat{Y}_F$ satisfies data minimization at level $\beta$ with $\alpha$ percent confidence if the probability that $\beta_j^b < \beta$ for at least one feature $f_j \in F$ and one imputation value $b \in \mathcal{X}_j$ is less than or equal to $1 - \alpha$.*

Intuitively, satisfying this guarantee at a high confidence $\alpha$ means that with high probability, there does not exist a simple feature imputation using which the model prediction changes for less than $\beta$ fraction of samples drawn from distribution $\mathcal{P}_\mathcal{X}$.

---

[1]We only consider the necessity of each individual feature for achieving system outputs, and leave the more general case of considering all feature combinations for future work.

**Probabilistic audit.** Now we present a mechanism for providing the above probabilistic guarantee. First, the probabilistic auditor can adopt a Bayesian approach to measure the uncertainty about the model instability under different simple imputations. Assuming a prior distribution for each $\beta_j^b$, the success probability of the Bernoulli variable $I_{\hat{Y}_F}(X, f_j, b)$, the auditor can apply the Bayesian update rule to achieve its posterior distribution based on the observations about the model instability under $\tau_{f_j, b}()$. Each observation corresponds to querying the prediction model for investigating the model instability under this imputation for a data point drawn randomly from $\mathcal{P}_{\mathcal{X}}$.

In particular, let $S_j^b$ denote the number of observations for which the simple imputation $\tau_{f_j, b}()$ changes the model prediction, and $F_j^b$ be the number of observations whose prediction does not change. Using the standard choice of modeling the mean of a Bernoulli variable with the Beta distribution and the resulting update rule, we get the posterior distribution $\beta_j^b \sim Beta(a + S_j^b, c + F_j^b)$ when the prior belief is $\beta_j^b \sim Beta(a, c)$ for each feature $f_j$ and imputation value $b \in \mathcal{X}_j$.

Next, we explain how an auditor can use the the posterior distributions of all $\beta_j^b$s to infer a probabilistic data minimization guarantee, i.e., verify whether with high probability $\alpha$, all $\beta_j^b$s are greater than some level $\beta$.

To satisfy the probabilistic data minimization guarantee at level $\beta$ with confidence $\alpha$, Definition 2 requires the probability of the following event to be small (less than or equal to $1 - \alpha$): *"for at least one feature $f_j \in F$ and one imputation value $b \in \mathcal{X}_j$, it is true that $\beta_j^b \leq \beta$."* Formally, we can rewrite the statement as:

$$Pr[\exists (f_j \in F, b \in \mathcal{X}_j) \text{ s.t. } \beta_j^b \leq \beta] \leq 1 - \alpha \tag{4}$$

We can use Boole's inequality to find an upper bound for the the probability in the left hand side of (4). We can find a lower bound for the same using the observation that the probability that $\beta_j^b \leq \beta$ for at least one feature $f_j$ and imputation $b$ is greater or equal than the probability that $\beta_j^b \leq \beta$ for any arbitrary $f_j$ and $b$. In particular we have:

$$\max_{f_j \in F, b \in \mathcal{X}_j} Pr[\beta_j^b \leq \beta] \leq Pr[\exists (f_j \in F, b \in \mathcal{X}_j) \text{ s.t. } \beta_j^b \leq \beta] \leq \sum_{f_j \in F, b \in \mathcal{X}_j} Pr[\beta_j^b \leq \beta] \tag{5}$$

Given the posterior distribution of $\beta_j^b$, let $L_j^b(\beta) = F_{Beta}(\beta; a + S_j^b, c + F_j^b)$ denote its cumulative distribution function. We can rewrite (5) as:

$$\max_{f_j \in F, b \in \mathcal{X}_j} L_j^b(\beta) \leq Pr[\exists (f_j \in F, b \in \mathcal{X}_j) \text{ s.t. } \beta_j^b \leq \beta] \leq \sum_{f_j \in F, b \in \mathcal{X}_j} L_j^b(\beta) \tag{6}$$

The auditor can use these bounds to decide whether or not a prediction system satisfies data minimization at *a given level $\beta$ and a given confidence level $\alpha$* as follows:

If the upper bound in (6) is less than or equal to $(1 - \alpha)$, then the auditor can declare that prediction system satisfies the probabilistic data minimization guarantee (as the probability that at least one $\beta_j^b$ is less than $\beta$ is smaller than $(1 - \alpha)$). On the other hand, if the lower bound in (6) is greater than or equal to $\alpha$, the auditor can declare that prediction system does not satisfy the probabilistic data minimization guarantee at level $\beta$ with confidence $\alpha$. If neither of the above conditions apply, the auditor cannot decide and would need to issue more queries to the black-box model and improve its estimate of posterior distributions of $\beta_j^b$s .

Alternately, if the auditor is concerned with measuring the best data minimization level that the prediction system satisfies with given confidence $\alpha$, they can leverage the observation that the upper bound in (6) is a monotonically increasing function of $\beta$, and apply a binary search to find $\beta$ such that the upper bound in (6) is equal to $1 - \alpha$. The resulting $\beta$ would be the best data minimization level that can be guaranteed with confidence $\alpha$ based on the posterior distributions[2].

## 5   Auditing With a Limited Query Budget

Using the framework introduced in Section 4, an auditor can provide a probabilistic data minimization guarantee based on the posterior distributions that are created for the model instability with respect

---

[2]A visualization of our probabilistic audit mechanism is provided in the supplementary (Appendix B).

to each simple imputation. As an auditor investigates the model instability with respect to different imputations, each posterior distribution is updated. Clearly, reducing the uncertainty about the instabilities of certain imputations may be more effective than others in finding a data minimization guarantee with high confidence.

Since it is desirable to limit or minimize the total number of queries used, a challenging problem faced by the auditor is how to distribute queries to investigate the effect of different simple imputations on the model instability. In particular, while a naive auditing approach may allocate equal numbers of queries for updating each posterior distribution, reducing the uncertainty about the model instability with respect to certain imputations might be more helpful than others, when providing a probabilistic data minimization guarantee. Thus an intelligent auditing strategy would spend more queries on investigating those imputations.

We address the above query allocation problem by introducing auditing algorithms that query the prediction system strategically. Hence we cast the problem of allocating a query budget to simple imputations into a bandit framework. Within this framework we consider two stopping criteria, corresponding to the two auditing task types. In particular, we formally define two bandit problems that correspond to the following tasks: (i) measuring the greatest data minimization level satisfied by a prediction model given a fixed query budget, and (ii) deciding whether or not data minimization is satisfied at a given level using the minimum number of system queries. In Section 6 we introduce auditing algorithms for each of the above problems.

**A Multi-armed Bandit Framework.** The multi-armed bandit problem [13] is a standard framework for modeling sequential decision problems under uncertainty, in which the actions (choices) are defined by a set of arms. A player sequentially chooses arms to play, and observes noisy signals of their quality, also known as rewards. The goal is then to optimize some utility while acquiring new knowledge about the arms. The query allocation problem for an auditor can be appropriately formulated as a stochastic bandit in which the rewards are modeled by a Bernoulli distribution associated with each arm.

In particular, we consider an arm for each pair $(f_j, b)$ of input feature $f_j \in F$ and feasible imputation value $b \in \mathcal{X}_j$. In each round, the auditor chooses an arm $(f_j, b)$ and observes a binary reward that is a sample from a Bernoulli distribution with success probability $\beta_j^b$. More specifically, each observation of arm $(f_j, b)$ corresponds to evaluating $I_{\hat{Y}_F}(X, f_j, b)$ at a data point $\mathbf{x}$ drawn randomly from $\mathcal{P}_\mathcal{X}$. This evaluation requires querying the prediction system to check whether the model prediction for $\mathbf{x}$ changes using the simple imputation associated with $(f_j, b)$. The success probabilities, i.e., the mean rewards of the arms, are unknown to the auditor. Furthermore, we incorporate the Bayesian assumption introduced in Section 4 and model the success probability of each arm using a Beta distribution whose shape parameters depend on the observations from that arm.

Bandit algorithms are traditionally developed for optimizing cumulative reward, a goal that requires both exploration and exploitation. However, our auditing problem is a *pure exploration* bandit since the auditor's objective is to explore arms in order to obtain a good estimation of the mean rewards (i.e., model instabilities). Previous approaches to pure exploration problems have focused on two main objectives: minimizing simple regret [4], and the best arm identification problem [1]. Our objective however is different from these, as we explore arms in order to provide a *probabilistic* data minimization guarantee. As explained in Section 4, equation (6) can be used to provide two types of guarantees depending on whether the data minimization level $\beta$ is fixed or not. Consequently, in the following we define two pure exploration bandit problems, where each is associated to a different auditing task: a decision problem, and a measurement problem. Both problems use the bandit setting described above, but each has a different set of input parameters and a different stopping condition, which results in a different data minimization guarantee.

**Decision Problem: Fixed Confidence and Fixed Level.** In this problem, the auditor's goal is to guarantee with a given confidence $\alpha$ that whether or not a prediction system satisfies data minimization at a given level $\beta$. A good auditing strategy for this problem tries to use a small number of system queries to provide this guarantee. Formally, based on equation (6) and using the bandit framework introduced in 5, the auditor seeks to solve the following problem:

*"Given confidence $\alpha$ and data minimization level $\beta$, iteratively select an arm $(f_j, b)$ to explore, and update the posterior distribution of $\beta_j^b$ based on the observed rewards; such that the resulting posteriors induce $\sum_{\substack{f_j \in F \\ b \in \mathcal{X}_j}} L_j^b(\beta) \leq (1 - \alpha)$ or $\alpha \leq \max_{\substack{f_j \in F \\ b \in \mathcal{X}_j}} L_j^b(\beta)$ using the minimum number of observations."*

**Measurement Problem: Fixed Confidence and Fixed Budget.** Alternatively, in this problem the auditor is given a fixed query budget and the goal is to measure $\beta$, the highest level of data minimization that the prediction system is guaranteed to satisfy, with a given confidence $\alpha$. Note that as explained in Section 4, for a fixed confidence $\alpha$ and at any state of posterior beliefs, $\beta$ can be computed using a binary search. A good auditing strategy however uses its query budget to provide the highest level of data minimization guarantee it can. Formally, we define the following bandit problem:

*"Given confidence $\alpha$ and query budget $T$, iteratively select an arm $(f_j, b)$ to explore, and update the posterior distribution of $\beta_j^b$ at each round; such that after $T$ rounds the value $\beta$ that satisfies $\sum_{f_j \in F, b \in \mathcal{X}_j} L_j^b(\beta) = (1 - \alpha)$ is maximized."*

## 6 Auditing Algorithms

In this section we present our algorithms for addressing the auditing problems defined in Section 5. These algorithms provide a framework for meeting the probabilistic guarantees required by the decision and measurement versions of a data minimization audit. Furthermore, we address the need to make audits efficient. The key challenge in making an audit efficient is to intelligently select which queries to present to the system under audit. We present a number of strategies, including novel strategies designed for our probabilistic setting; we will demonstrate the efficiency of these strategies in Section 7.

Algorithms 1 and 2 present solution strategies for the two versions of the auditing problem. Each algorithm is assumed to be able to query the prediction model $Y_F$, and to be able to sample an audit dataset $\mathcal{D}$. Algorithm 1 summarizes the auditing procedure for solving the decision problem with confidence $\alpha$ and data minimization level $\beta$, where $\alpha$ and $\beta$ are provided as the inputs to the algorithm. The auditing procedure for solving the measurement problem is presented in Algorithm 2. In this algorithm instead of fixing the data minimization level, a fixed query budget $T$ is given as the input and the algorithm returns the level of data minimization that can be guaranteed to be satisfied with confidence $\alpha$.

---

**Algorithm 1:** Data Minimization Audit (Decision Problem)

**Input:** $Y_F$, $\mathcal{D}$, confidence $\alpha$, level $\beta$

1   $S_j^b \leftarrow 0$, $F_j^b \leftarrow 0$; $(\forall f_j \in F \; \forall b \in \mathcal{X}_j)$
2   **repeat**
3     $(f_j, b) \leftarrow \texttt{SelectArm}(\beta)$
4     Draw $\mathbf{x}$ from $\mathcal{D}$ uniformly at random
5     Query $Y_F$ to evaluate $r = I_{Y_F}(\mathbf{x}, f_j, b)$
6     Increment either $S_j^b$ or $F_j^b$ based on $r$
7   **until** *A decision can be made for $\alpha$ and $\beta$*;
8   **return** *The binary decision made using Eq.(6)*

---

**Algorithm 2:** Data Minimization Audit (Measurement Problem)

**Input:** $Y_F$, $\mathcal{D}$, confidence $\alpha$, budget $T$

1   $S_j^b \leftarrow 0$, $F_j^b \leftarrow 0$; $(\forall f_j \in F \; \forall b \in \mathcal{X}_j)$
2   **for** $t$ *in 1 to $T$* **do**
3     Find $\beta^*$ using a binary search
4     $(f_j, b) \leftarrow \texttt{SelectArm}(\beta^*)$
5     Draw $\mathbf{x}$ from $\mathcal{D}$ uniformly at random
6     Query $Y_F$ to evaluate $r = I_{Y_F}(\mathbf{x}, f_j, b)$
7     Increment either $S_j^b$ or $F_j^b$ based on $r$
8   **return** $\beta^*$

---

Each algorithm has at its core an exploration strategy, i.e., a decision about the next query to present to the system, which we denote $\texttt{SelectArm()}$ (reflecting the bandit problem viewpoint). This decision uses the current knowledge about the reward distributions of the arms, and some exploration strategies use a given level $\beta$ as well. As described in the previous section, the choice of which bandit arm to sample corresponds to a choice of $(f_j, b)$ and the sampling itself consists of querying $I_{Y_F}(\mathbf{x}, f_j, b)$ where $\mathbf{x}$ is a random data point from $\mathcal{D}$. The algorithms accumulate knowledge about an arm $(f_j, b)$ by maintaining success and failure counters $S_j^b$ and $F_j^b$.

When solving the decision problem (Algorithm 1), the arms are explored iteratively until a decision can be made based on the posterior distributions of the success rates of arms. That is, until the auditor

can accept or reject that the prediction model satisfies data minimization at level $\beta$ with confidence $\alpha$ using Eq.(6). For the measurement problem on the other hand (Algorithm 2), the auditing algorithm keeps selecting arms and querying the system until all the query budget is used. The largest data minimization level that can be guaranteed with confidence $\alpha$ is then computed using a binary search.

Now we present the strategies for choosing which imputation to apply to the query sample used in each iteration of the auditing algorithms. While the bandit setting is a natural one for our problems, the probabilistic nature of our problems does not correspond to any classical bandit problem. Hence, we propose and evaluate heuristic exploration strategies that are either based on classical approaches or new approaches we design in light of the specific nature of our problems. In particular, we propose four exploration strategies: the first two algorithms are based on Thompson Sampling, while the second two are designed specifically for obtaining a lower bound guarantee on the mean reward[3].

**Thompson Sampling (TS).** Our first approach is based on Thompson sampling [22]. Thompson sampling is a heuristic for maximizing the expected reward when choosing actions sequentially under uncertainty, and it is shown to have good performance in practice [5]. We adapt Thompson sampling to be used in our auditing algorithms as an arm selection procedure. In particular, at each iteration and for each arm $(f_j, b)$ a sample $\theta_j^b$ is drawn from $Beta(x; S_j^b + a, F_j^b + c)$ given the current success and failure counters. The arm that corresponds to the *minimum* sample is then selected to be explored next. The idea behind Thomson sampling is to explore an arm according to the chance that reducing the uncertainty about its mean reward would better help finding a data minimization guarantee, which is a probabilistic lower bound on all success probabilities. Thus, we choose the arm that corresponds to the minimum sample, instead of the maximum sample used in the original Thompson sampling.

**Top-Two Thompson Sampling (TTTS).** While Thompson sampling is effective in maximizing the expected cumulative reward, it is not a good strategy if the goal is to identify the optimal arm, as it may select a suboptimal arm at the outset, and then continue exploring that arm, resulting in little chance for other arms to be explored. In fact, it is known that algorithms achieving small cumulative regret cannot be optimal for the best arm identification problem [4]. Top-Two TS [20] addresses this drawback by randomly choosing between two of the best alternatives at each iteration. That is, with probability $\gamma$ it returns the arm selected by TS, and with probability $1 - \gamma$ it keeps sampling until an alternative arm is selected.

Previous work has studied the sample complexity of the best arm identification problem [4] as well as the convergence rate of TS-based strategies for both the expected cumulative regret [21] and the best arm identification problem [20]. In particular, it has been shown that using Top-Two TS, the probability that a sub-optimal arm is selected converges to zero at an exponential rate [20]. Notice that although identifying the arm with minimum mean reward induces a lower bound on all mean rewards, our problem is, in some sense, easier than the best arm identification problem since it does not require exact identification of the best arm as far as a probabilistic lower bound on all arms is achieved.

**Greedy.** The previous two algorithms use the posterior beliefs about the mean rewards of the arms, and select arms with respect to their probability of having the minimum mean reward. However, the data minimization guarantee that we seek depends on the probability mass that is below some threshold $\beta$ in all arms. Based on this observation, we develop two new approaches.

In the first approach, instead of sampling the posterior distributions at each iteration, we first evaluate $L_j^b(\beta)$, the cumulative distribution function at $\beta$, for all arms. Then, using a greedy approach, we select the arm whose posterior beta distribution has the maximum probability mass below $\beta$. Given that our data minimization guarantee for both decision and measurement problems depends on either the total sum or the maximum value over all $L_j^b(\beta)$'s, the greedy selection more closely matches our algorithmic goal.

**Probability Matching using CDFs (PM).** Our second new algorithm adopts the probability matching strategy of TS, and applies it to the cumulative distribution functions of arm rewards at the given threshold $\beta$. That is, it selects arms in proportion to the amount of probability mass that is below $\beta$ in each reward distribution, thus allowing more arms to be explored compared to Greedy.

---

[3]Pseudocode for these algorithms are provided in the supplementary (Appendix C).

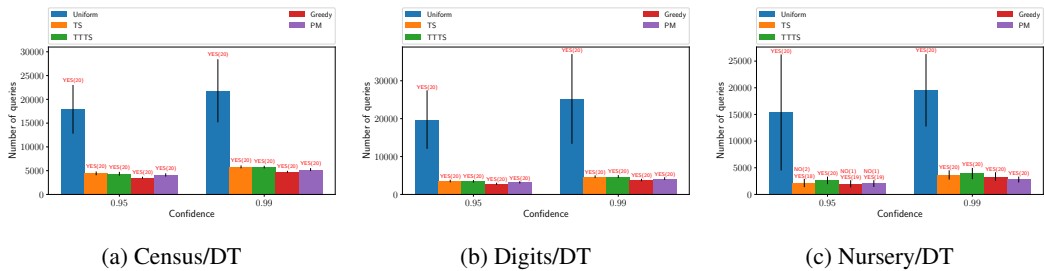

Figure 1: Auditing for $1\%$ data minimization.

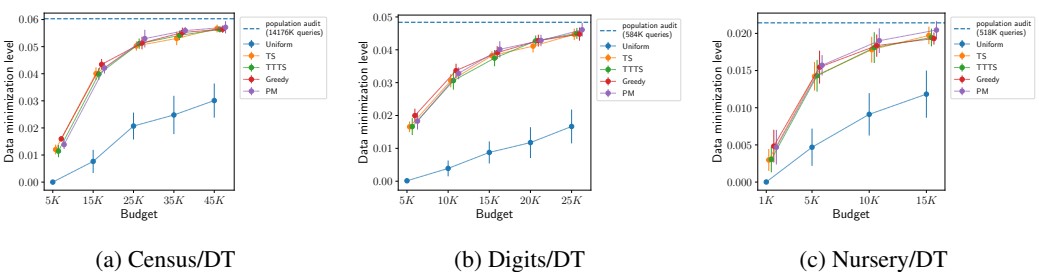

Figure 2: Measuring data minimization with $95\%$ confidence.

# 7 Auditing Real-world Prediction Systems

In this section we study the effectiveness of the algorithms introduced in section 6 in auditing real-world prediction systems for data minimization compliance[4].

We build prediction systems using datasets from the UCI machine learning repository [7]. We use three datasets that have discrete features and discrete target variables, and we apply different learning algorithms together with standard feature selection and model validation methods to build a prediction model. In particular, we use the following prediction systems.

**Digits/SVM.** A dataset of 3823 images of hand-written digits from the MNIST [14] database is used. Each data point is an $8 \times 8$ matrix whose elements are integers from the range $[0, 16]$, along with a label from the set $\{0, 1, \dots, 9\}$. A support vector machine with linear kernel is used to build a classifier that predicts the label associated with each image. $50\%$ of data points are used to train the classifier, and a recursive feature elimination procedure is applied to select 9 features.

**Census/Decision Tree.** A dataset of $\sim 30K$ individuals from the US Census database is used. Eleven features with a discrete domain are available for each individual, and the goal is to predict whether a person makes over $\$50K$ a year. A decision tree is built using $20\%$ of data samples as the training data, and a recursive feature elimination procedure is applied to select 5 features.

**Nursery/Decision Tree.** This dataset contains 8 discrete features from $\sim 13K$ applicants for nursery schools in Slovenia. A label from a set of 5 priority groups is assigned to each applicant, and we use $20\%$ of data samples to build a decision tree for predicting the priority group of each applicant.

In the following experiments, for each system we use the whole data as the audit dataset from which query samples are drawn. All prior distributions for $\beta$ parameters are $Beta(1/2, 1/2)$ (i.e., the Jeffreys prior). The decision parameter $\gamma$ in TTTS strategy is set to $1/2$.

In our first experiment, we apply Algorithm 1 to decide whether each of the above systems satisfy data minimization at $1\%$ level. We perform this task for 0.95 and 0.99 confidences, and using the exploration strategies introduced in section 6. As a baseline algorithm, we also implement a uniform exploration strategy that investigates all (feature, imputation value) pairs uniformly until a decision can be made.

---

[4]Code is available at: `https://github.com/rastegarpanah/Data-Minimization-Auditor`

The bar chart in Figure 1 shows the average number of system queries used by each exploration algorithm and for each confidence level over 20 runs. The error bars show one standard deviation and the resulting Yes/No decisions, together with the frequencies of decisions, are printed over each bar. We observe that all three systems satisfy data minimization at $1\%$ level, and our exploration algorithms reach this decision using significantly fewer queries compared to the uniform exploration method. Furthermore, the algorithms designed specifically for our auditing purpose (`Greedy` and `PM`) are on average slightly more effective than standard Thomson sampling based algorithms [5].

Next we apply Algorithm 2 to measure the level of data minimization satisfied by each system using different query budgets. We consider a .095 confidence for this measurement task, and in order to have a comparison point if the were no limit on the query budget, we compute the probabilistic data minimization level using the population audit, i.e., using all the data points in the audit dataset.

Figure 2 shows the level of data minimization that each algorithm can guarantee for each system and using different query budgets. In each experiment we increase the budget until one of the algorithms reaches the level measured by population audit. We observe that all of our exploration strategies significantly outperform the uniform exploration, i.e., they can guarantee a higher level of data minimization at the given confidence for each query budget, and they can reach the level guaranteed by population audit using less than $5\%$ of queries used by population audit. Finally, we observe that for smaller query budgets, `Greedy` and `PM` are on average more effective than TS-based algorithms.

Lastly, we illustrate the performance of our auditing framework for a hypothetical scenario in which a data minimization audit may be particularly needed. We consider a case in which a prediction system has an input that is not actually used by the prediction model, i.e., the system collects excessive information from its users. We obtain such system by embedding the Census/DT model in another prediction system that asks for an additional attribute of the Census dataset but returns the output of the embedded model.

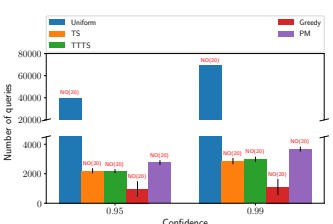

Figure 3

We apply the decision auditing algorithm for $1\%$ data minimization level to this systems. Figure 3 presents the number of queries used, and the output of the auditing algorithms for each exploration strategy and two confidence levels. We observe that all algorithms return "No", i.e., detect that data minimization is not satisfied, as desired. More importantly, the exploration algorithms we propose use an order of magnitude less queries to make this decision. And we particularly note that our `Greedy` strategy is significantly more effective than the other exploration strategies considered.

## 8   Summary and Concluding Remarks

In this paper we provide an operationalization of data minimization principle that is applicable to auditing prediction models. We propose using model instability as the purpose for which data needs to be minimized and we suggest exploiting simple imputations as a tool for limiting data inputs at the test time. Adopting a Bayesian approach and a bandit framework, we provide efficient auditing algorithms that measure model instability with respect to different simple imputations and provide a probabilistic data minimization guarantee.

We initiated the audit framework assuming discrete feature values. In the scenarios where feature domains are relatively large (e.g., integers), more efficient bandit algorithms can be developed using ideas similar to [26, 11, 12], or [25] for infinitive domains. Other directions for future research include extending our individual features guarantee to the case of all feature combinations, and studying the implications of instability-based data minimization on different fairness notions similar to works that have done it for accuracy-based data minimization [18].

Finally, although the framework we use in this paper is agnostic to the implementation details of the prediction model, an open question concerns the impact on data minimization of different feature selection techniques employed during model construction. On the other hand, our model-instability metric itself suggests a feature importance metric, which may be used for feature selection; similar instability-based feature importance metrics have been previously proposed (e.g., [6]).

---

[5]The few "No"s in Figure 1c correspond to the probabilistic nature of our guarantee.

## Acknowledgments

This research was supported by the ERC Advanced Grant "Foundations for Fair Social Computing" (No. 789373), and the National Science Foundation under grant numbers IIS-1421759 and CNS-1618207. We thank the anonymous reviewers for their helpful comments, which greatly improved the presentation of this paper.

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
