# Auditing Black-Box Prediction Models for Data Minimization Compliance (Supplementary Material)

**Bashir Rastegarpanah**
Boston University
bashir@bu.edu

**Krishna P. Gummadi**
MPI-SWS
gummadi@mpi-sws.org

**Mark Crovella**
Boston University
crovella@bu.edu

## A    Setting

Figure 1 demonstrates our setting for auditing black-box prediction models as described in Section 3.

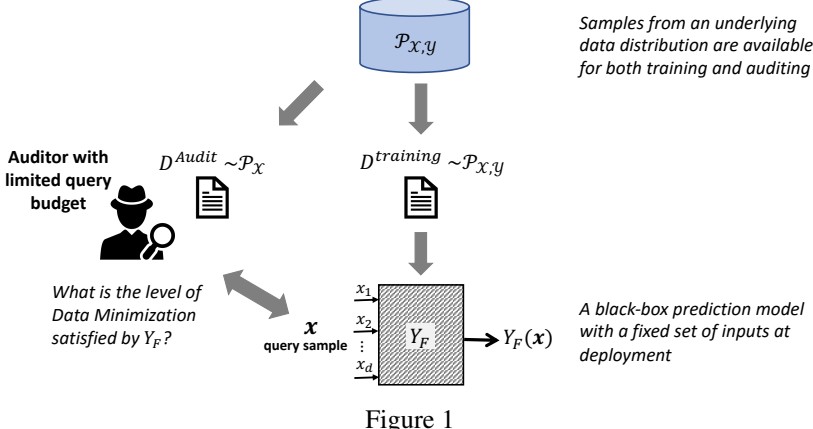

Figure 1

# B Probabilistic Audit

Figure 2 demonstrates examples of how the posterior distributions of model instability with respect to different simple imputations can be used to derive a probabilistic data minimization guarantee.

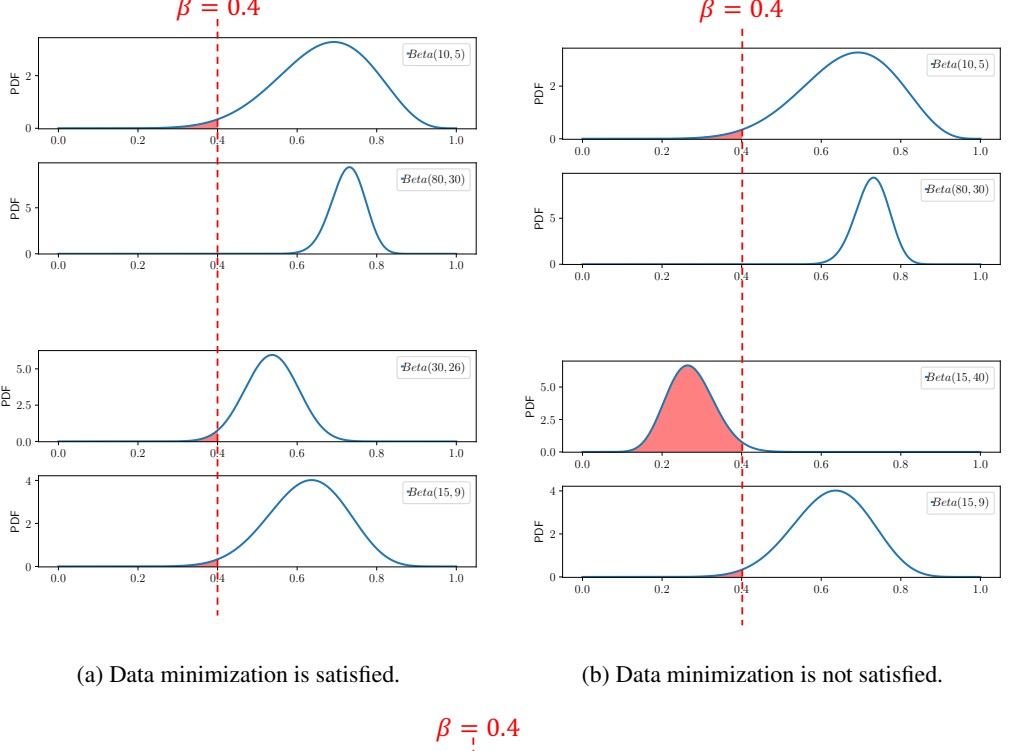

(a) Data minimization is satisfied.

(b) Data minimization is not satisfied.

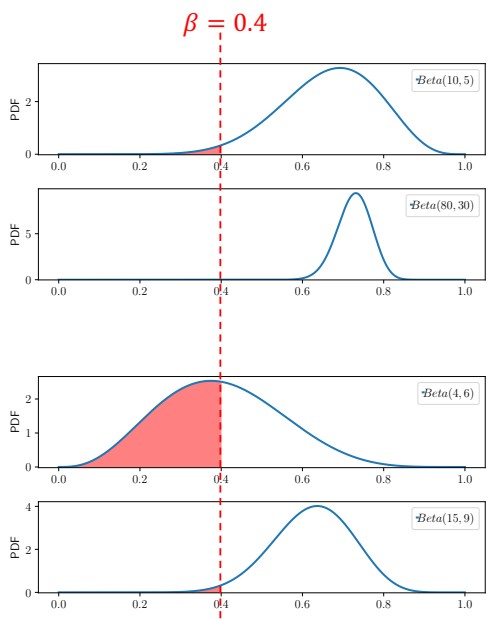

(c) An example of a situation where a decision cannot
be made based on the posterior distributions.

Figure 2: Example posterior distributions of model instability with respect to different imputations, and a probabilistic data minimization audit at level $0.4$ with $90\%$ confidence.

## C Pseudocodes for Exploration Strategies

---

**Algorithm 1:** Thompson Sampling (TS)

---

**Input:** Success and failure counters

1 **for** $f_j \in F$ **do**
2     **for** $b$ *in* $\mathcal{X}_j$ **do**
3         $\theta_j^b \sim Beta(x; S_j^b + a, F_j^b + c)$
4 $j^*, b^* = \operatorname{argmin}_{j,b} \theta_j^b$
5 **return** $(f_{j^*}, b^*)$

---

**Algorithm 2:** Top-Two Thompson Sampling (TTTS)

---

**Input:** Success and failure counters

1 **for** $f_j \in F$ **do**
2     **for** $b$ *in* $\mathcal{X}_j$ **do**
3         $\theta_j^b \sim Beta(x; S_j^b + a, F_j^b + c)$
4 $j^*, b^* = \operatorname{argmin}_{j,b} \theta_j^b$
5 $K \sim Bernoulli(1/2)$
6 **if** *K=1* **then**
7     **return** $(f_{j^*}, b^*)$
8 **else**
9     **repeat**
10         **for** $f_j \in F$ **do**
11             **for** $b$ *in* $\mathcal{X}_j$ **do**
12                 $\theta_j^b \sim Beta(x; S_j^b + a, F_j^b + c)$
13         $\tilde{j}, \tilde{b} = \operatorname{argmin}_{j,b} \theta_j^b$
14     **until** $(\tilde{j}, \tilde{b}) \neq (j^*, b^*)$;
15     **return** $(f_{\tilde{j}}, \tilde{b})$

---

**Algorithm 3:** Greedy

---

**Input:** Success and failure counters, $\beta^*$

1 **for** $f_j \in F$ **do**
2     **for** $b$ *in* $\mathcal{X}_j$ **do**
3         $p_j^b = F_{Beta}(\beta^*; S_j^b + a, F_j^b + c)$
4 $j^*, b^* = \operatorname{argmax}_{j,b} p_j^b$
5 **return** $(f_{j^*}, b^*)$

---

**Algorithm 4:** Probability Matching (PM)

---

**Input:** Success and failure counters, $\beta^*$

1 **for** $f_j \in F$ **do**
2     **for** $b$ *in* $\mathcal{X}_j$ **do**
3         $p_j^b = F_{Beta}(\beta^*; S_j^b + a, F_j^b + c)$
4 Randomly choose an arm $(f_j, b)$ with probability $\dfrac{p_j^b}{\sum\limits_{(f_j, b)} p_j^b}$
5 **return** $(f_j, b)$

---