# OpenReview forum: "Auditing Black-Box Prediction Models for Data Minimization Compliance"
_NeurIPS.cc/2021/Conference — NeurIPS 2021 Spotlight_

### Official Review · Reviewer_xkzE · 2021-07-14

**Rating:** 7
**Confidence:** 2

**Summary:**

The authors explore means for the blackbox auditing predictive models in line with data minimisation concerns. The key focus is on the role of particular features, i.e. whether they are necessary for the model or not.

Towards this, the paper describes an instability-based data minimisation metric, based on various feature imputations, as well as a probabilistic measure of uncertainty (termed a data minimisation guarantee).

Given imputations can be expensive (require many queries), approaches considering query-budgets are described, where bandit frameworks are used for considering the greatest data minimisation level satisfied given a fixed query-budget, and whether a particular level of minimisation is satisfied using the minimum number of queries.

The approach and efficiencies of the query budget are demonstrated across three real-world datasets.

**Ethical Concerns:**

Fairly minimal. Though the paper should acknowledge that tackling issues of data minimisation are beyond what's discussed here.

**Limitations And Societal Impact:**

There isn't really any discussion of the limitations in the paper.

Some of that raised in the main review point to some of these.  I think it's important that the work is properly placed in its broader context -- that such approaches won't *solve* data minimisation problems, but might help.

**Main Review:**

I found this paper relevant and interesting. Data minimisation is an important, and topical problem, and more work on this is certainly welcome.

Metrics and methods regarding the impact of features in this context have the potential to encourage actions towards less 'invasive' systems (from a personal data perspective). Having means to assist minimisation, through defining probabilistic measures/metrics, and showing ways for reducing the associated overheads are of relevance to the community. Intuitively the methods seem sound, but I defer to the other reviewers to assess the specifics.

The evaluation appeared largely to focus on performance and the number of queries. This is clearly important. However, I was anticipating more about where data minimisation requirements are *not* met, and the role and effectiveness of the methods in uncovering these. This is done very briefly in Figure 3, in a scenario where some extraneous input features are added that do not impact the model. But there is a real opportunity for more exploration unpacking situations that are more messy, involving more features, etc (even if these are hypothetical/synthetic). (And indeed, something could have been said about the few "NO"s that cropped up in Fig 1). Given the general aim and context of this work, I think some exploration regarding scenarios that differ in how much they violate 'minimisation' is warranted.

The paper is generally well written. But I think there are some framing and contextual points that would better place the work.

First regards the practicalities of the context and scenarios in which the use of these methods are envisaged. That is, how do they fit wrt a system? Given the focus is a 'black-box' audit, would this be e.g. through a remote service? These sorts of questions matter - e.g. to help understand if the method reducing 20k queries to 3k is sufficient? Moreover, many deployed systems won't have direct model interactions, but the model will operate as part of a larger system. This means a bunch of other data might be inputted to support the business process, only some of which flow to the model. Are there any implications of this?

Related is the intended target for these methods. The paper talks about an auditor, and with the legal discussion this comes across as a regulator. Would a regulator really employ a blackbox approach vs. using their other powers to audit and assess? In reading the paper, I actually felt this method might be better aimed at *internal* auditors (e.g. by developers/Q&A testers) rather 'external' entities. Some discussion clarifying things would help.

More broadly, it should be explicit that such approaches won't solve data minimisation concerns. GDPR is about technical *and* organisational measures - while what goes into a model can help, what is appropriate or not will depend on the specifics of the broader socio-technical processes involved.

Note that these latter comments don't detract from the approach itself, but some mention of these sorts of things can help better frame the paper, the contributions and generally strengthen things.

**Time Spent Reviewing:**

3

---

> ### Author Response · Authors · 2021-08-10
> **We explain how we will use the reviewer's comments to improve the paper.**
>
> We appreciate the helpful comments made by the reviewer and we will use them to improve the paper.
>
> We agree that exploring more cases that unpack situations in which data minimization is not satisfied is interesting. Given the limited space we had, we initiated exploiting this question using the experiment presented in Figure 3, and we would like to explore more complex situations in follow on work.
>
> The few “No”s in Fig. 1 correspond to the probabilistic nature of our guarantee and we will clarify this in the camera-ready version.
>
> When reducing the number of system queries, our assumption is that it is natural to have a limited number of system queries due to privacy and performance issues (e.g. network overhead). While the bandit problem defined in the paper (decision problem) asks for using the minimum number of queries to reach a decision, we believe the significant effectiveness (using an order of magnitude less queries) of our auditing strategies compared to simpler benchmarks is of practical importance.
>
> Addressing the intended target of our framework, our mental model has been an external third-party auditor who does not have access to model internals. However, since we are concerned with efficient auditing algorithms, we agree that our auditing tools might be of interest to an internal tester as well, and will add these points to the paper.
>
> Finally, we thank the reviewer for highlighting the important fact that our approach does not solve the general data minimization problem in its full social context. Rather, we are proposing tools that help address some aspects of the bigger data minimization problem. We will add acknowledgment of this limitation to the paper.

---

### Official Review · Reviewer_73BZ · 2021-07-15

**Rating:** 6
**Confidence:** 4

**Summary:**

This paper presents a framework for auditing black-box prediction models for compliance with GDPR's data minimization principle. The audit framework is based on checking the necessity of each individual feature used in the prediction models by imputing them with constant values and checking the extent of variation in the predictions.

**Limitations And Societal Impact:**

The limitations with respect to applicability for automatic feature detectors (e.g. deep neural nets) are not addressed.

---- after rebuttal ---
The above concern has been addressed by the authors.

**Main Review:**

The key contributions of the paper are:
1. formalizing the setting for auditing models with respect to the data minimization principle
2. developing algorithms for how the necessity for individual features can be assessed through imputations with a limited query budget.
3. providing a metric for model uncertainty under feature imputations.

These are important contributions for machine learning models where the feature set is specified apriori.

My main concerns are regarding the practical relevance of the proposed framework. Since the framework needs to impute each feature specified apriori individually to determine their necessity, automated feature detection based predictors (like deep neural networks) are not applicable for audit by the framework. This is a major limitation of the framework that prevents applicability on real-world datasets and complex models beyond SVMs and Decision Trees evaluated in the experiments.

**Time Spent Reviewing:**

4

---

> ### Author Response · Authors · 2021-08-10
> **We explain how deep neural networks would be audited using our framework.**
>
> We thank the reviewer for bringing up the important case of automated feature detection based predictors, and we will add the discussion of how such models (e.g., deep neural networks) would be audited under our framework.
>
> A DNN whose initial layers learn a set of representations, still requires an input layer that consists of one node for each original feature. This input layer can be thought of as the interface available to the auditor, which contains a fixed set of input variables. Alternatively, a DNN may be used to find representations based on which a prediction model is built. Given that we are concerned with auditing a prediction model at the deployment time, our framework can be applied to the input layer of this DNN which is part of the prediction model pipeline. Basically, although a DNN will learn representations internally, or implicitly select features internally, it can still be treated as a black box to the external auditor. We believe that in this way, the audit process can be applied to DNNs as well. Finally it is worth mentioning that in this paper we assume the prediction system itself is a privacy adversary.

---

> > ### Comment · Reviewer_73BZ · 2021-08-20
> > **Thank you for the clarification**
> >
> > Thank you for clarifying how the framework applies to DNNs. I will revise my score based on this clarification.

---

### Official Review · Reviewer_jdGr · 2021-07-21

**Rating:** 8
**Confidence:** 4

**Summary:**

This paper defines a new setting generalizing existing work on establishing _data minimization_ as motivated by the GDPR. Unlike previous work, this work considers testing for data minimization in a black-box setting (previous work assumes access to model internals). Within this setting, the data minimization problem is defined according to a measure of model instability when certain features are replaced by constant imputation (as is often done for missing values in data preprocessing). Further, the problem of auditing a model for data minimization at a certain level (according to a metric the paper introduces) is framed as a pair of near-dual bandit problems: one which establishes the optimal auditing strategy to achieve a certain confidence that the metric is not violated, and another which assigns a fixed query budget in an expected-optimal way.

**Ethical Concerns:**

No ethical concerns.

**Limitations And Societal Impact:**

The work aims to address societal impact of other machine learning activities by providing insight into the extent to which a model relies on a minimal set of features. However, as noted in the main review, I think the work would be strengthened by more explicitly acknowledging limitations like explaining the settings in which claims of the sort the proffered technique produces will be acceptable/useful or acknowledging that collected data might be used in secondary places not examined by this approach or discussing the efficiency of the sampling strategies proposed more completely.

**Main Review:**

I enjoyed this paper. I believe most of the things that don't quite work are also limitations of the prior work on which it builds, so I elect not to hold them against the paper. However, I will describe them in case that aids in the presentation.

First, a black-box auditing setting is most useful for outsiders like activists or journalists. Enforcement entities or internal control entities would have the ability to demand access to model internals, and would want that anyway. Also, a probabilistic guarantee is less suitable in these cases, but is probably acceptable to outsiders who can't get details but might be able to interact with a tool in a query-oriented way. The paper doesn't say this explicitly and I think it should - in places it even intimates that the technique would be useful for enforcement-type or control-type auditing, and it should avoid that. That is, the value of the auditing tools to a specific constituency/application could be more clearly motivated, especially given the probability-oriented metric around which the contribution centers. Relatedly, there should be an explicit claim about whether the data considered for "data minimization" has any secondary uses beyond training the model under evaluation. One might fairly exclude such secondary use based on the way the GDPR ties data minimization explicitly to purpose limitation, but that may go too far into legal analysis of unsettled questions, risking that the paper's motivation would end up on the "wrong side" of future jurisprudence and guidance.

Second, I would appreciate some more careful contextualization of these new measures into existing literature on feature importance measures and feature engineering more broadly - if a feature is unnecessary in a data minimization sense, why should it be included at all? A typical answer would say that it helps to differentiate hard cases where other features coincide, so (as is noted in the discussion) there could be real and explicit tradeoffs with performance on minority subgroups or identification of rare phenomena. It would be interesting to explore this. I'm not deeply familiar with methods in the feature importance world, so it's hard for me to say to what extent the framing here is novel. But the setting is certainly novel, so I'm happy to leave that as an open question for other reviewers to fill in.

Third, the algorithms supplied to solve the two bandit problems look correct to me, but even in the extensive discussion of sampling strategies there is no mention of the worst-case or even average-case sample complexity/convergence. It would be interesting to at least qualify this beyond the experiments performed (which seem to indicate that convergence is rapid enough to perform many runs without difficulty) - for example, is complexity dependent on the distribution in the problem being examined? Does it require distributional assumptions? How can an auditor know if convergence in a reasonable number of samples is likely?

Some tiny points not worth including in the review narrative:
* The paper claims to "introduc[e] the notion of a probabilistic audit" (at 155, but maybe elsewhere?). But this is not new. Even in financial auditing, it is quite common to use a sampling methodology to identify transactions in a record or activities within a corporation to reconcile against governance processes and then to express the audit's findings in terms of any identified deficiencies as well as a confidence level achieved that unexamined activities will not reveal additional deficiencies. So this claim should be weakened.

**Time Spent Reviewing:**

2

---

> ### Author Response · Authors · 2021-08-10
> **We explain how we will incorporate the reviewer's comments about the 3 points that will improve the presentation of the paper.**
>
> We thank the reviewer for suggestions for improving the presentation of the paper. In the following we detail how we will incorporate the reviewer’s helpful comments in the camera-ready version.
>
> We agree and will clarify that our framework is intended for outsiders like activists or journalists, rather than entities who have full access to system internals (i.e. prediction algorithm). More generally, the black-box setting and the query-oriented audit is useful for a third party who does not have access to the prediction model internals at the level that enforcement entities or internal control entities would have. Given that prediction algorithms are often an important business asset, it is reasonable to consider auditing mechanisms in a black-box setting. Furthermore, as it is mentioned by the reviewer, if some features are gathered for secondary use (e.g. to prevent racial discrimination), those features can be specified a priori and excluded from the auditing procedure. We will include these points in our revision.
>
> The contextualization of our metrics into existing literature on feature importance and feature engineering would definitely improve the paper. We were aware of this and we will discuss this issue in more detail given the 1 page additional space which is available if the paper is accepted. As mentioned by the reviewer, features might be added to the model for identification of rare phenomena or performance on minority subgroups. Also it is helpful to think about the measures introduced in this paper independently of the feature selection procedure. For instance, as it is demonstrated in Fig. 3, it can even be the case that a system collects excessive information intentionally, which is an interesting case for auditing data minimization compliance. However, it is indeed an interesting research question to study how various feature selection techniques affect our data minimization metric. In our experiments, we had applied standard backward elimination feature selection methods in order to build the example systems. Alternatively, one can study the effect on model instability of other feature selection methods. Finally, our model-instability metric itself suggests a feature importance metric that might be useful for feature selection too. Similar instability-based feature importance metrics are previously proposed (eg Datta et. al. AAAI 2015). However, to the best of our knowledge, the distributional model-instability metric and the idea of using a lower bound on the instabilities with respect to all possible simple imputations are novel.
>
> We will address the questions regarding the convergence of sampling strategies in the paper revision. From a high level, the sampling strategies are designed to reduce the uncertainty about the mean rewards such that a probabilistic lower bound on all mean rewards can be achieved. While this bound can be always inferred if the posterior beliefs are concentrated enough around the true mean of each arm, the sampling strategies focus on exploring arms that updating their posterior mean distribution better helps finding a lower bound and stop exploring arms whose mean is unlikely to be close to the minimum instability. For this task, a problem instance in which multiple arms have a mean reward close to the minimum instability of all arms is harder to solve. However our problem is easier than the best arm identification problem in the sense the exact identification of the best arm is not required as far as a probabilistic lower bound on all arms can be achieved. The sample complexity of the best arm identification problem is previously studied (eg Bubeck et. al., ALT 2009), and the convergence rate of TS-based strategies are studied for both the expected cumulative regret (Russo, Van Roy 2014) and the best arm identification problem (Russo, COLT 2016). In particular it is shown that using the Top-Two TS, the probability that a sub-optimal arm is selected converges to zero at an exponential rate (Russo, 2016). Intuitively, identifying the arm with minimum mean reward with high confidence induces a probabilistic lower bound on all mean rewards. Finally, note that the measurement algorithm (Algorithm 2) is applicable given any query budget, i.e., for any status of posterior beliefs. Therefore, if the decision algorithm cannot make a decision using some given query budget, Algorithm 2 can be applied to find the largest data minimization level that the auditor can guarantee.

---

### Decision · Program_Chairs · 2021-09-27

**Decision:**

Accept (Spotlight)

**Comment:**

The authors of this paper propose a principled approach to evaluate the data minimization principle of a black box model. Based on a defined model instability metric, the authors detail in their contribution how to audit a black box model (several sampling strategy are proposed) and how to assess how unstable the system becomes after several imputations. Real applications on small datasets are provided at the end of the paper.

The paper is very well written and clear. The problem tackled is highly relevant. While the probabilistic audit si well known and not new, the authors embed it nicely in their task at hand, auditing a black box model to assess the data minimization guarantee. Invoking multi arm bandit in their method is clever. This paper is an operational paper with a clear detailed method and some illustrations.

Based on the reviews and my own reading, I would like to suggest the authors to consider the following items, to further improve the quality of their paper:

1) more extensive experiments.
2) guarantees on the data minimization principle depending on how one audits and how large the dataset is.
3) discuss how the imputation (single imputation is used here) strategy can impact the stability of the model.